# Oncolytic Viral Therapy for Glioma by Recombinant Sindbis Virus

**DOI:** 10.3390/cancers15194738

**Published:** 2023-09-27

**Authors:** Kangyixin Sun, Xiangwei Shi, Li Li, Xiupeng Nie, Lin Xu, Fan Jia, Fuqiang Xu

**Affiliations:** 1Wuhan National Laboratory for Optoelectronics, Huazhong University of Science and Technology, Wuhan 430074, China; sunkangyixin@hust.edu.cn; 2Shenzhen Key Laboratory of Viral Vectors for Biomedicine, Key Laboratory of Quality Control Technology for Virus-Based Therapeutics, Guangdong Provincial Medical Products Administration, NMPA Key Laboratory for Research and Evaluation of Viral Vector Technology in Cell and Gene Therapy Medicinal Products, The Brain Cognition and Brain Disease Institute (BCBDI), Shenzhen Institute of Advanced Technology, Chinese Academy of Sciences, Shenzhen-Hong Kong Institute of Brain Science-Shenzhen Fundamental Research Institutions, Shenzhen 518055, China; shixiangwei17@mails.ucas.ac.cn (X.S.); l.li1@siat.ac.cn (L.L.); 3State Key Laboratory of Magnetic Resonance and Atomic and Molecular Physics, Key Laboratory of Magnetic Resonance in Biological Systems, Wuhan Center for Magnetic Resonance, Innovation Academy for Precision Measurement Science and Technology, Chinese Academy of Sciences, Wuhan 430071, China; 4University of Chinese Academy of Sciences, Beijing 100049, China; 5CAS Key Laboratory of Animal Models and Human Disease Mechanisms, KIZ-SU Joint Laboratory of Animal Model and Drug Development, Laboratory of Learning and Memory, Kunming Institute of Zoology, Chinese Academy of Sciences, Kunming 650201, China; nie_xiupeng@gzlab.ac.cn (X.N.); lxu@mail.kiz.ac.cn (L.X.); 6Center for Excellence in Brain Science and Intelligence Technology, Chinese Academy of Sciences, Shanghai 200031, China

**Keywords:** glioblastoma, oncolytic virus, sindbis virus, cytokines, killing glioma

## Abstract

**Simple Summary:**

Gliomas account for 27% of all primary central nervous system tumors and have a highly aggressive growth pattern, leading most patients to be insensitive to conventional treatment. Viotherapy as a progressive therapeutic tool may be beneficial to improve the outcome of brain tumor treatment. However, the viruses used for glioma treatment at this stage are mostly limited to a few engineered replication-free defective viruses. In this study, we provide a replication-competent sindbis virus as a glioma treatment that, when combined with cytokines, is effective in slowing glioma progression both by intratumoral injection and systemic administration. This provides a promising concept for the treatment of glioma and also provides valuable data on the safety of using replicable sindbis virus as a glioma-killing agent.

**Abstract:**

Background: The characteristics of glioblastoma, such as drug resistance during treatment, short patient survival, and high recurrence rates, have made patients with glioblastoma more likely to benefit from oncolytic therapy. Methods: In this study, we investigated the safety of the sindbis virus by injecting virus intravenously and intracranially in mice and evaluated the therapeutic effect of the virus carrying different combinations of IL-12, IL-7, and GM-CSF on glioma in a glioma-bearing mouse model. Results: SINV was autologously eliminated from the serum and organs as well as from neural networks after entering mice. Furthermore, SINV was restricted to the injection site in the tree shrew brain and did not spread throughout the whole brain. In addition, we found that SINV-induced apoptosis in conjunction with the stimulation of the immune system by tumor-killing cytokines substantially suppressed tumor development. It is worth mentioning that SINV carrying IL-7 and IL-12 had the most notable glioma-killing effect. Furthermore, in an intracranial glioma model, SINV containing IL-7 and IL-12 effectively prolonged the survival time of mice and inhibited glioma progression. Conclusions: These results suggest that SINV has a significant safety profile as an oncolytic virus and that combining SINV with cytokines is an efficient treatment option for malignant gliomas.

## 1. Introduction

Oncolytic virus therapy is a promising cancer treatment because it not only kills tumor cells directly but also serves as a vehicle for heterologous genes to drive immune system activation within the tumor. Since the FDA approved the first oncolytic virus therapy in 2015 [1], various oncolytic viruses that selectively target and kill specific cancer cells for killing have been reported, and most of these studies have focused on adenoviruses, herpesviruses, and poxviruses. 

The sindbis virus (SINV), a member of the genus Alphavirus in the family *Togaviridae*, was isolated from *Culex* mosquitoes near Cairo in 1952 [2]. It is a zoonotic virus that causes symptoms such as arthritis, fever, and rash in the early stages of infection, and symptoms of arthritis may persist for months or years after infection [3,4]. SINV is a positive single-stranded RNA virus that encodes four non-structural proteins (NSP1-4) and five structural proteins (C, E3, E2, 6K and E1), of which the E2 glycoprotein is the virulence protein of the virus, and the amino acid mutation in E2 affects the neurotoxicity of the virus [5,6,7]. At this stage, there have been many studies on the structure and function of SINV, as well as the development of SINV vectors to provide new options for the delivery of target proteins and gene therapy [8,9].

SINV has the following advantages as an oncolytic virus: it is a blood-borne virus and can therefore reach most tissues of the body, and the entire RNA genome of SINV is about 12,000 nucleotides, which makes it easy to engineer [10,11]. Moreover, as an alphavirus, SINV avoids the risk of genomic integration and the induction of cell apoptosis [11,12,13]. Moreover, SINV induces apoptosis with the help of the 67 kDa high-affinity laminin receptor (LAMR) [12,14]. The 67 kDa high-affinity laminin receptor is frequently overexpressed in tumor cells compared to normal cells and correlates with the metastatic and invasive capacity of tumors [15,16,17], which allows SINV to efficiently kill cancer cells. During the infection of host cells, SINV triggers an antitumor immune response in vivo by increasing the number of CD8+ T cells and NK cells [18,19]. Many studies have reported the ability of replicable SINV to efficiently kill different tumors, such as pancreatic, ovarian, and cervical cancers [20,21,22]. In contrast, there are few studies related to the killing of glioblastoma by SINV.

Glioblastoma multiforme (GBM) is the most aggressive glioma currently known, and it is difficult to treat via surgical resection; GBM patients have a median overall survival (OS) of only 15 months [23]. At present, several clinical trials using oncolytic viruses are underway or have been performed in patients with GBM [24,25,26]. The neurotoxicity caused by viruses during treatment is a significant obstacle for the oncolytic virus therapy of GBM, and the viruses commonly used to treat GBM, such as herpes simplex virus (HSV) or vesicular stomatitis virus (VSV), are neurotoxic to humans, which makes these wild-type viruses unsuitable for clinical treatment [27,28]. Nevertheless, engineered HSV and VSV can be used as glioma-killing agents, for example, by knocking ICP34.5 out of HSV or VSV ΔM51 [29,30]. Of concern is that in contrast to wild-type VSV, SINV can infect neurons in mice without killing them, and the virus progressively vanishes over time [31]. Therefore, we hypothesize that replication-competent SINV may be a novel therapeutic agent for the treatment of GBM.

## 2. Materials and Methods

### 2.1. Cell Lines

BHK-21 (RRID: CVCL_1914) and HEK293T (RRID: CVCL_0063) cells were obtained from the American Type Culture Collection (ATCC, Manassas, VA, USA). U-87MG (RRID: CVCL_0022), BV-2 (RRID: CVCL_0182), HMC3 (RRID: CVCL_ll76), U-118MG (RRID: CVCL_0633), C6 (RRID: CVCL_0194), and GL261 (RRID: CVCL_Y003) cells were obtained from Procell Life Science&Technology Company (Wuhan, China). BHK-21, HEK293T, U-251MG, U-118MG, A-172, and GL261 cells were maintained in Dulbecco’s modified eagle medium (DMEM, Thermo Fisher, Waltham, MA, USA) supplemented with 10% Fetal Bovine Serum (FBS, Thermo Fisher) and 1% penicillin-streptomycin (Thermo Fisher); U-87MG, BV-2 and HMC3 cells were maintained in Eagle’s Minimum Essential Medium (Thermo Fisher) supplemented with 10% FBS and 1% P/S; C6 cells were maintained in Ham’s F-12 K Medium (Thermo Fisher) with 15% horse serum and 2.5% FBS and 1% P/S.

Construction of a U-87MG-Luc cells line: The firefly luciferase gene and the enhanced green fluorescent protein (EGFP) gene were inserted into a lentiviral vector and expression was driven with a murine cytomegalovirus promoter. Afterward, the engineered lentiviral vector was transfected on HEK293T cells and the lentiviral particles were collected after 72 h post-transfection. The obtained lentiviral supernatant was repeatedly infected with U-87MG cells until all cells expressed green fluorescence. Next, The EGFP-positive cells, that is, cells expressing luciferase in U87-MG cells were sorted out by flow cytometry and continued in culture.

### 2.2. Viruses and Titration

The plasmids of pSINV-EGFP were the same as those we used in previous research [31], which have an EGFP driven by a 26 s promoter after the E1 protein. Granulocyte-macrophage colony-stimulating factor (GM-CSF) (GenBank: EU366957.1), interleukin-7 (IL-7) (GenBank: J04156.1), and interleukin-12 (IL-12) (GenBank: AF101062.1) were inserted into SINV-EGFP in place of EGFP using two restriction enzyme sites, AscI and NotI, respectively. Further, the SINV expressing IL-12/IL-7; IL-12/GM-CSF; IL-7/GM-CSF was constructed based on SINV vectors with a single inserted cytokine. Similarly, the other protein was inserted into the non-structural protein of the virus via the restriction enzyme sites FseI and PmeI between the Nsp4 and Cap proteins. All plasmids were verified by DNA sequence.

The viral vector inserted with cytokines was transfected into BHK-21 cells that had been inoculated one day in advance using lipo2000, and the viral supernatant was collected 48 h later and cell debris was removed by passing it through a 0.22 µm filter membrane. Afterwards, BHK-21 cells in 10 cm dishes were continued to be infected with 1 µL of viral supernatant to achieve viral amplification, and the viral supernatant was collected 48 h after infection and centrifuged at 2000× *g* for 5 min. After a sufficient amount of viral supernatant was collected, centrifuge for 2.5 h at 50,000× *g* in a high-speed centrifuge. Next, the virus precipitate was resuspended with 1 mL of PBS, after which 20% sucrose solution and the resuspended virus solution were added to the centrifuge tubes in a 5:1 volume ratio, and the same was centrifuged at 50,000× *g* for 2.5 h. At the end of centrifugation, the viral precipitate was resuspended with 50 µL of PBS and dispensed into PCR tubes. The virus was stored at −80 °C and repeated freeze–thaw was avoided.

Viral titers were determined by the standard plaque assay. Briefly, virus titers were calculated from the number of plaque-forming units after gradient dilution of the viral solution infecting BHK-21 cells.

### 2.3. Cell Viability Assay

U-87MG, BV-2, HMC3, U-118MG, C6, and GL261 cells were inoculated in 96-well plates one day in advance with 4 × 10^4^, 4 × 10^4^, 6 × 10^4^, 6 × 10^4^, 5 × 10^4^, and 8 × 10^4^, respectively. Different types of cells were then infected with SINV-EGFP in the different multiplicity of infection (MOI). After 48 h, the dead/live cells were identified using the Calcein/PI Cell Viability Assay Kit (Beyotime, Shanghai, China), where the live cells were stained with Calcein and labeled with green fluorescent markers, and the red fluorescent markers were PI-stained dead cells. Live cells were counted by hemacytometer in each well, and cell viability was calculated.

### 2.4. Safety Assessment of SINV

For the safety evaluation, C57BL/6 mice were injected intraperitoneally with 100 microliters (µL) (1 × 10^7^ plaque-forming units (PFU) per dose) of SINV or vehicle twice, every other day. Sixty days later, the vital organs (brain, heart, kidney, liver, lung, spleen, and muscle) of mice were taken for subsequent H&E-stained and histological analysis. Similarly, 100 µL of SINV (1 × 10^7^ PFU) was injected intravenously with C57 mice and BHK-21 cells were infected by 100 µL blood sample collected intravenously from mice hourly. Afterwards, total RNA was extracted from the heart, head, spleen, liver, kidney, and blood of mice collected at 1, 5, and 10 days post-injection using the MiniBEST Universal RNA Extraction Kit (Takara, Shiga, Japan), and viral RNA was converted to cDNA using the PrimeScript RT Master Mix (Takara). Next, cDNA was treated by Tip Green qPCR Supermix (Bio-red, Hercules, CA, USA), and relative cDNA levels were calculated by comparing the Ct (cycling threshold) method. This enables the quantification of viral RNA in these organs to obtain results on the biodistribution of systemically delivered SINV.

A total of 0.1 µL (1.0 × 10^5^ PFU) of SINV-EGFP was injected into the caudate putamen (CPu) region of the nude mouse, and the brain was taken after 6 and 16 days. The brain was placed in 4% paraformaldehyde (PFA) for 48 h, then replaced with 30% sucrose until it sank to the bottom and was followed by sectioning. Similarly, SINV was injected into the superior colliculus (SC) of the tree shrew, and 7 days later, the brain was taken by perfusion.

### 2.5. Animal Models

This study was approved by the Animal Care and Use Committees at Innovation Academy for Precision Measurement Science and Technology, the Chinese Academy of Sciences (approval No. APM22026A) in 2022. C57BL/6 mice and Balb/c nude mice were provided by Hunan SJA Laboratory Animal Co., Ltd., Changsha, China (license No. SCXK (Xiang) 2019-0004). All experiments on the tree shrews (*Tupaia belangeri*) were approved by the Experimental Animal Core Facility of the Kunming Institute of Zoology (KIZ), Chinese Academy of Sciences (CAS). All animals were kept in a standard environment, weighed regularly, and euthanized when they lost more than 20% of their body weight or when the tumor diameter exceeded 20 mm.

For the establishment of the subcutaneous tumor model of glioblastoma, 5 × 10^6^ U-87MG cells with PBS or 100 µL Ceturegel^®^ Matrix LDEV-Free Matrigel (Yeasen, Shanghai, China) were transplanted into the right posterior thigh of 4-week-old nude mice. After 7 days, the mice were randomly divided into different groups (n = 5 for each group) and injected with 100 µL (1 × 10^6^ PFU per dose) of SINV or phosphate-buffered saline (PBS), three times, every other day. The length and width of the tumor were measured every other day, and the volume of the tumor was calculated according to the formula 1/2 (length × width^2^). Further, intracranial models of glioma were established: 5 × 10^5^ U-87MG-Luc cells were injected into the right striatum of a 6-week-old nude mouse by using stereotaxic apparatus. Seven days post-cell implantation, mice were randomly injected with 3 µL (1 × 10^6^ PFU) of SINV IL-12/IL-7 or 3 µL PBS into the tumor. Tumors were measured with the IVIS imaging system at 28 days post-injection (dpi); during this process, we closely observed the weight change and abnormal behavior of mice.

### 2.6. IVIS Imaging

To observe tumor growth in living mice, luciferase images were obtained using the Small Animal In Vivo Imager system to track luciferase expressed by tumor cells. D-Luciferin Potassium Salt (Yeasen, Shanghai, China) was injected intraperitoneally into mice at a concentration of 150 milligram/kilogram (luciferin/body weight) 10 min before imaging and the Living image version 4.2 was used to quantify the expression of luciferase.

### 2.7. Flow Cytometry Analysis

After 4 days of SINV injection into the U-87MG subcutaneous tumor model, tumor tissue was taken in a 100 mm petri dish, and each sample was treated separately as follows: rinsed with 1× HBSS and then transferred to a 15 milliliter (mL) centrifuge tube after the tissue had been cut up with scissors. A total of 5 mL of 10× triple enzyme mix (1 g Collagenase IV, 100 mg Hyaluronidase, and 20,000 units DNase in 100 mL HBSS solution, filtered through a 0.22 micrometer filter, dispensed, and stored at −20 °C; all enzymes used were obtained from Yeasen) was added to the centrifuge tube and digested overnight at room temperature. The digestion was then suspended by adding RPMI 1640 medium containing serum (Thermo Fisher) and centrifuged at 300× *g* for 5 min. The precipitate was resuspended in 100 µL PBS to make a cell suspension, to which 2 µL of primary antibody was added and incubated on ice in the dark for 20 min. Cells were resuspended by adding 200 µL PBS, and 1 µL Zombie Aqua™ Fixable Viability Kit (Cat# 423101, Biolegend, San Diego, CA, USA) was added to exclude dead cells, incubated on ice for 3–5 min, and then analyzed by flow cytometry. The antibodies used in the analysis were as follows: APC anti-mouse CD3 (Cat#100236, Biolegend); PE anti-mouse CD49b (Cat#103506, Biolegend). The results obtained were analyzed using Flowjo, version 10.8.1 software.

### 2.8. Statistical Analysis

All data are presented as mean ± S.E.M., and GraphPad Prism 8.0 was used for processing all graphs and statistical analysis of the data. The Kaplan–Meier method and the log-rank test was used for the analysis of survival rates. *p* < 0.05 was considered statistically different, *p* < 0.01 was considered statistically significant, and *p* < 0.001 was considered extremely statistically different.

## 3. Results

### 3.1. SINV Is Safe for Mice

Safety is a prerequisite for a drug to be used in clinical treatment. Therefore, we evaluated the safety of SINV in animals. SINV-EGFP (1 × 10^7^ PFU) was administered intravenously to C57BL/6 mice. Blood samples were taken from the mice every hour after injection, and the samples were used to infect BHK-21 cells. Fluorescence signals were observed after 24 h (Figure 1A) and the viral titer within the serum was measured using a plaque assay (Figure 1B). After 5 h post-administration, SINV was barely detectable in the blood, suggesting that SINV disappeared after a short stay in the blood following systemic administration. Furthermore, the biodistribution of viral RNA in vital tissues of mice was quantified via quantitative RT-PCR (qRT-PCR) (Figure 1C). These results showed a transient enrichment of SINV in the organs after systemic administration, with a significant reduction in the organs after 5 days post-injection (dpi) and a returned to normal values at around 10 days. H&E staining results from approximately 60 days post-injection showed that SINV also did not cause histological changes in the liver, muscle, brain, lung, or heart, but minor structural abnormalities exist in the kidney and the spleen (Figure 1D). Overall, SINV, as replicable oncolytic virus, is a good candidate for tumor killing. Further, 0.1 µL (1.0 × 10^5^ PFU) of SINV-EGFP was injected into the right caudate putamen (CPu) of the nude mice. The fluorescence signals were observed at the injection site and in the primary motor cortex (M1), a brain region adjacent to the CPu. Significant fluorescence signals were also observed in the dorsal endopiriform nucleus (DEn) and the ventral posteromedial thalamic nucleus (VPM) areas downstream of CPu on day 6, indicating that the signal had propagated to the next secondary neuron (Figure 1E). Notably, the fluorescence signal decreased substantially in all brain regions as the time increased; no significant signal was observed in the DEn and VPM brain regions at 16 dpi, but a faint signal and apoptosis of neurons caused by SINV were observed at the injection site (Figure 1E). Additionally, to study the neurotoxicity of SINV in primate-liked animals, we injected SINV-EGFP into the superior colliculus (SC) of the tree shrew (*Tupaia belangeri*). The EGFP signal was observed only at the injection site at 7 dpi, with no signal observed in other brain regions (Figure 1F). These results further show that SINV is safe for use in the brain in mice and tree shrews.

### 3.2. SINV Can Kill GBM Cells In Vitro

To determine the susceptibility of GBM cell lines to SINV, U-87MG, U-118MG, HMC3, BV-2, GL261, and C6 cells were infected with SINV-EGFP at MOIs of 1, 0.1, and 0.01. Intense green fluorescence expression was observed in the U-118MG, C6, and U-87MG cells after 24 h post-infection, whereas no green signal, due to widespread SINV infection, was observed in the other cells (Figure 2A). This suggests that SINV is able to selectively infect some glioma cell lines, such as U-87MG, C6, and U-118MG cells, without infecting GL261 and normal glial cells (BV-2 and HMC3). Therefore, the U-87MG cells were chosen for the subsequent construction of the tumor-bearing model. The results of cell viability assays showed that SINV-EGFP was caused significant cell death in a dose-dependent manner (Figure 2B). In contrast, for insensitive cells, SINV caused little apoptosis in these cell lines, even at high infective doses (MOI = 1). These data suggest that SINV can selectively infect a subset of glioma cell lines in vitro and effectively induce apoptosis in the infected cells.

### 3.3. SINV Effectively Kills U-87MG Subcutaneous Tumors

Next, the ability of SINV to kill tumors in vivo was evaluated. U-87MG subcutaneous tumors were established in the nude mice by injecting 5 × 10^6^ U-87MG cells into the subcutaneous tissue near the right thigh. Afterwards, the mice received intratumoral injections of 1 × 10^6^ PFU of SINV or PBS only on 7, 9, and 11 days after tumor implantation (Figure 3A). Changes in tumor volume were monitored after injection. It was found that the tumor volume in the PBS-injected mice was notably larger than that in the SINV treatment group (Figure 3B). During treatment, continuous SINV injection did not cause weight loss or other adverse effects in the immunodeficient nude mice (Figure 3B). In addition, H&E staining of tumor tissues from the PBS-injected mice showed vigorous growth of tumor cells in a tight arrangement, whereas the tumor tissues from the SINV-treated mice showed obvious necrosis of the tumor cells (Figure 3C). These results demonstrate that SINV is capable of retarding tumor growth in vivo and is a promising tumor-killing virus for GBM.

### 3.4. Cytokines Improve the Tumor-Killing Ability of SINV

Previous studies showed that cytokines can improve the efficacy of oncolytic virus in killing tumors. [32,33] Here, we selected three cytokines (IL-7, IL-12, and GM-CSF) to assess their ability for enhancing the efficacy of SINV in eradicating tumors. Cytokines were inserted into SINV vectors singly or in two-by-two combinations, with insertion sites as shown in Figure 4A. The viability of BHK-21 and U-87MG cells that had been exposed to several recombinant SINVs containing various cytokines was first assessed. After 24 h post-infection, all the recombinant viruses induced more cell death in the U-87MG cells than in the wild-type SINV, whereas the recombinant SINVs induced less apoptosis in BHK-21 cells. The numbers of apoptotic cells were proportional to the dose of viral infection (Figure 4B). Next, these recombinant SINVs were intratumorally injected into the subcutaneous U-87MG tumor in animal models at 7, 9, and 11 dpi to assess their tumor-killing ability in vivo (Figure 4C). The change in tumor volume and weight indicated that the combinations of two cytokines had superior tumor-killing efficacy to the single cytokines (Figure 4D,E). Collectively, the combination of IL-12 and IL-7 had greater efficacy than the other combinations of cytokines. In addition, flow cytometry analysis indicated that treatment with cytokine-armed SINVs increased the proportion of NK cells in the tumors of mice treated with cytokine-carrying SINVs compared to the SINV-treated groups (Figure 4F). Taken together, these results show that cytokines can notably enhance the tumor-killing ability of SINVs.

### 3.5. SINV IL-12/IL-7 Treatment Prolongs Survival in an Intracranial Glioblastoma Model

To more realistically simulate the growth environment of GBM in vivo, we established an orthotopic GBM model. In order to better observe tumor growth in the mice, we constructed a U-87MG cell line stably expressing luciferase (U-87MG-Luc). At 7 days after U-87MG-Luc cells were implanted into the CPu of the mice, the mice received intratumoral injections of SINV IL-12/IL-7 or PBS, followed by animal imaging at 21 dpi to evaluate tumor growth (Figure 5A). The SINV IL-12/IL-7-treated groups exhibited weaker luciferase signals compared to the control groups, which indicated that SINV can significantly reduce tumor size (Figure 5B,C). In addition, Kaplan–Meier survival curves showed that SINV IL-12/IL-7 treatment significantly prolonged the mean survival time in the GBM mouse model (Figure 5D). It is worth highlighting that a proportion of mice in the SINV IL-12/IL-7 treatment group (approximately 80%) survived for more than 30 days without any detectable symptoms. In addition, brain tissue from the mice was taken at 21 and 45 dpi for histopathological examination. The H&E staining results showed that the brain tissue in the control group was abnormal, with large tumors visible, whereas SINV IL-12/IL-7 treatment effectively inhibited the growth of intracranial gliomas (Figure 5E). Overall, SINV IL-12/IL-7 exhibited significant GBM therapeutic efficacy in both the subcutaneous and intracranial tumor models.

### 3.6. Systemic Administration of SINV Can Kill GMB Transplanted in Peripheral Tissues

The above results suggest that SINV is highly effective in killing gliomas. Extracranial metastases have been reported to occur in 0.4–0.5% of all glioblastomas, with the common sites of metastasis including the lymph nodes, lungs, and bone [34,35]. Furthermore, the above results also show that SINV has a good safety profile after systemic administration. Therefore, we evaluated the killing effect of recombinant SINV on distal tumors following systemic administration. Firstly, we established a subcutaneous tumor model in mice using the U-87MG-Luc cell line by randomly grouping the mice and intravenously injecting them with 100 µL of SINV IL-12/IL-7 (1.0 × 10^7^ PFU) on days 7, 9, and 11 after cell transplantation (Figure 6A). At 7 days after tumor implantation, the tumor size, indicated by bioluminescence measurements, was uniform in all groups, while the treated group had fewer visible tumor masses and only mild luciferase signals compared to the control group at 21 dpi (Figure 6B–D). Similarly, results are observed from the time–tumor volume plots (Figure 6E), which indicates that SINV causes a significant regression in tumor growth following systemic administration. Therefore, SINV might be a promising candidate for tumor killing when GMB metastasizes from the brain to other organs.

## 4. Discussion

For glioblastoma (GBM), which is the most common and devastating primary tumor in adults, virotherapy is emerging as an approach for the treatment of GBM [36,37]. In this study, we evaluated the safety and efficacy of SINV as a treatment for GBM. It was determined that SINV was effective in inhibiting the growth of highly aggressive GBM tumor models induced by U-87MG cells, either via intratumoral injection or via systemic perfusion. In addition, the efficiency of the three cytokines (IL-12, IL-7, and GM-CSF) combined with SINV was also evaluated, and the results demonstrated that SINVs containing these cytokines were effective in promoting antitumor adaptive immunity and enhancing antitumor efficacy. This study suggests that SINV has the potential to be used in treatment for refractory intracranial malignancies.

To date, a variety of studies have reported the tumor-killing capacity of SINV [19,38,39]. However, few reports have systematically evaluated the safety of replication-competent SINV as an anti-cancer candidate virus [40,41]. Here, we discuss the safety of SINV after systemic administration. We determined that SINV was rapidly cleared from the bloodstream; at 1 h after intravenous injection, the virus titer in the blood was around 10^3^ PFU compared to the initial number of virus particles injected (1 × 10^7^ PFU), and the virus was almost completely cleared after 5 h (Figure 1B). This property of SINV results in low viremia and thus lower mortality [40]. Additionally, after intravenous injection, some of the previously described alphaviruses, including the Venezuelan equine encephalitis (VEE) virus, showed similar organ distribution and virus accumulation in the liver and spleen [40,42]. It has also been suggested that the ability of SINV to adhere to heparan sulfate (HS) correlates with circulating clearance [43,44]. Since the liver contains a large amount of highly sulfated HS [45], most of the protein removed from the circulation can be found in this organ [46,47]. However, the virus, which is concentrated in the liver, is completely cleared after 6 days post-injection and does not cause serious damage to the aggregation site [48]. The focus on the treatment of gliomas has led to the need for confirmation of the intracranial manifestation of SINV. We found that SINV infects neurons in adult mice and does not kill the mice in the process [31]. After 16 days post-injection, the infectious virus is cleared from the mouse brain. Previous studies have shown that the clearance of infectious viruses from neurons is associated with the secretion of gamma interferon (IFN-γ) by T cells and the production of anti-E2 glycoprotein antibodies (Abs) by B cells [41,49]. It has been shown that defective HSV, which is used to treat gliomas, replicates in cancer cells and causes cytopathic lesions [29]. This is similar to how replicable SINV can quickly reproduce in tumor cells before being eliminated in the circulation. Although DNA viruses such as HSV have multiple clinical applications for tumor killing, some RNA viruses have low pre-existing immunity in humans and are more suitable for systemic administration [50,51]. In addition, RNA viruses are more efficient in replication, leading to greater amplification inside the tumor. Furthermore, the Protein kinase R (PKR) pathway is strongly activated after cell infection by SINV [52], whereas tumor cells are often defective in their PKR signaling pathway [50], which provides broader application prospects for SINV as a tumor-killing agent. Overall, SINV provides an attractive platform from which to construct genetically engineered viruses with strong tumor-killing activity.

At this stage, the whole genome of the tree shrew has been sequenced, and because the neural, immune, and other systems of this species are similar to those of humans [53], this has led to the tree shrew being used in studies of different disease models, including depression [54], cancer [55], and viral infections [56]. In our work, the susceptibility to sindbis virus was also studied in the nervous system of the tree shrew. Unlike in mice, SINV does not propagate synaptically in the neural network of the tree shrew, which leads us to speculate that SINV would behave similarly in the primates. This study’s result provides a positive indication that SINV, as an oncolytic virus, may not spread to other brain regions beyond the injection site and can thus ensure the safety of the drug. Moreover, different strains of SINV behave very differently within their respective genera; the more neurovirulent strains (NSV and SVN) can induce severe encephalitis, leading to death in adult mice [7]. A similar situation is observed with other alphaviruses, such as the Semliki Forest virus (SFV), where the more neurotoxic SFV L10 can spread throughout the brain and cause fatal encephalitis, whereas SFV A7(74) is confined to the initial area of infection and is then cleared by the immune system [57]. These results suggest that the safety profile of different strains of SINV may vary and that further evaluation of the safety of SINV in primates is needed in the future.

SINV is naturally effective in inducing apoptosis in cancer cells, but it has previously been reported that treatment with wild-type SINV elicits only a small immune-related response [58]. The combination of SINV with immunostimulatory molecules is more effective in establishing a pathway for an effective immune response [39,58]. In this study, we evaluated the effects of several common cytokines in combination with SINV and found that arming either IL-7 or IL-12 increased the tumor-killing efficiency of SINV. Recombinant human IL-7 has been reported to increase CD4+ and CD8+ T cells in the periphery of cancer patients [59], while the expression of IL-12 in melanoma patients also increases the infiltration of CD8+ cytotoxic T cells into tumors [33]. Our data suggest a superior performance for SINV that co-carries both cytokines. Tumor-selective vaccinia viruses encoding IL-7 and IL-12 have previously been reported [32]; however, the short replication cycle of vaccinia viruses makes it difficult, in some cases, for them to secrete sufficient inflammatory cytokines to inhibit tumor growth prior to viral elimination. In contrast, alphaviruses are more persistent in the blood. This is important for the systemic administration of drugs targeting distant or metastatic tumors. However, if SINV is used in combination with cytokines in metastatic tumors, further data are needed to avoid the problems of the cytokine storm that may occur in patients receiving high-dose administration and the production of antibodies following repeated dosing.

## 5. Conclusions

In conclusion, this study provides evidence that SINV can safely and efficiently treat gliomas. Cytokine-armed SINVs triggered effective alterations in the tumor microenvironment in a mouse glioma model, stimulating adaptive immunity and thus enhancing the antitumor activity of SINV. These properties give us reason to believe that the possession of replicative activity while ensuring safety could make SINV a potential cancer therapy.

## Figures and Tables

**Figure 1 cancers-15-04738-f001:**
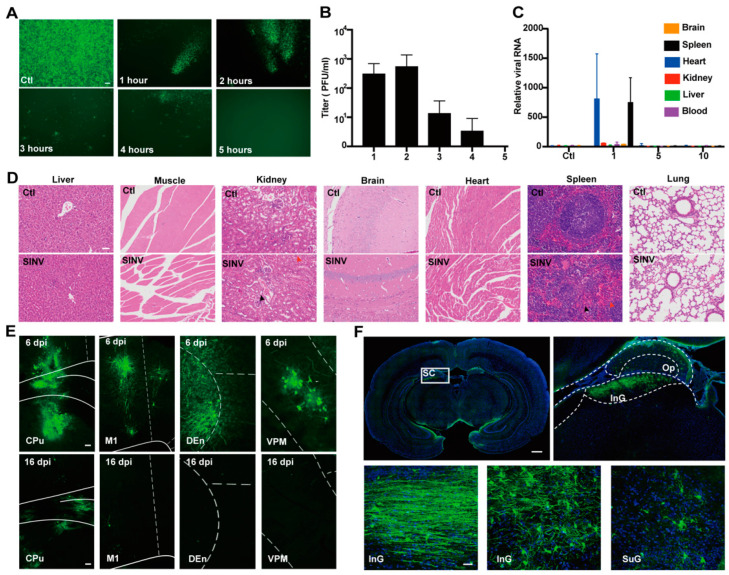
Safety assessment of SINV. (**A**) The 100 µL blood samples of mice from 1 to 5 h after tail vein injection of SINV-EGFP were collected and used to infect BHK-21 cells, areas with viral expression were selected for recording in an inverted microscope at 24 h post-infection. Scale bars, 100 µm; The mean value of the number of viral particles per 1 mL of blood samples is shown in (**B**). (**C**) The distribution of viral RNA extracted from the heart, liver, spleen, kidney, brain, and blood of mice after 1, 5, and 10 days (n = 3 at each time point) of intraperitoneal injection of SINV-EGFP. (**D**) H&E staining results of the organs of the mice at 60 days post-injection. As shown by the red arrow in the figure, the number of lymphocytes was reduced in the spleen, and more multinucleated giant cell infiltration was seen in the tissue, as shown by the black arrows in the Figure. In addition, a small number of loose and edematous renal tubular epithelial cells were seen in the kidney, as shown by the red arrow, and a small number of inflammatory cells were seen in the tissue, as shown by the black arrow. The rest of the tissue is normal and unlabeled. Scale bars, 100 µm. (**E**) The signal of the virus in different brain regions at 6 days as well as at 16 days after SINV intracranial injection (magnification, 10×). Scale bars, 20 µm. (**F**) The SINV could infect the neurons of tree shrews. The superficial gray layer, optic nerve layer, and intermediate gray layer of the superior colliculus were detected with green positive signals. Scale bars: upper left image, 1 mm; top right image, 200 µm; bottom image, 20 µm.

**Figure 2 cancers-15-04738-f002:**
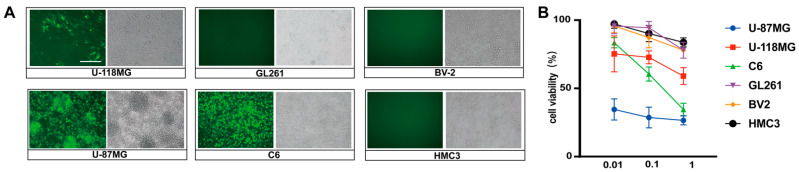
SINV-EGFP effectively induces apoptosis in some GBM cells in vitro. (**A**) Glioma cells and glial cells infected with SINV-EGFP (MOI = 0.1); representative images were obtained under an inverted microscope at 24 h post-infection (magnification, 20×). Scale bars, 100 µm. (**B**) All cells were infected with SINV-EGFP (MOI = 1, 0.1 or 0.01) and cell viability was assessed at 48 h post-infection. Cell viability was considered to be 100% in mock-infected cells. Data are presented as the mean and s.d. of 6 independent experiments.

**Figure 3 cancers-15-04738-f003:**
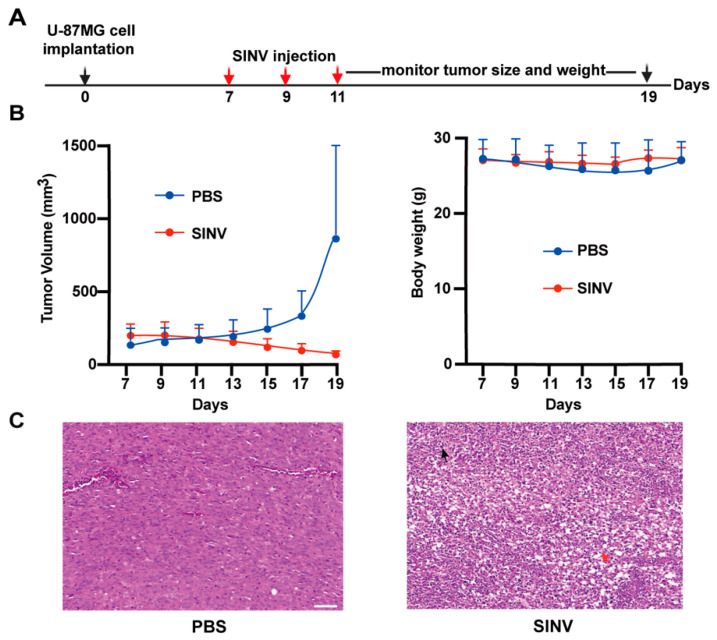
Tumor-killing effect of SINV on subcutaneous gliomas. (**A**) Treatment schema. After subcutaneous injection of U-87MG cells with PBS into the right thigh of nude mice, SINV-EGFP was injected into glioma tumors on days 7, 9, and 11 after cell implantation. (**B**) Tumor length and width were measured using calipers and tumor volume was calculated using the formula (n = 5 for each group), and the weight of the mice was also recorded continuously using an electronic scale. (**C**) H&E staining results of subcutaneous tumor in mice. The SINV group showed a large area of tumor cell necrosis, as shown by the red arrow, and a small amount of inflammatory cell infiltration, as shown by the black arrow. Scale bars, 50 µm.

**Figure 4 cancers-15-04738-f004:**
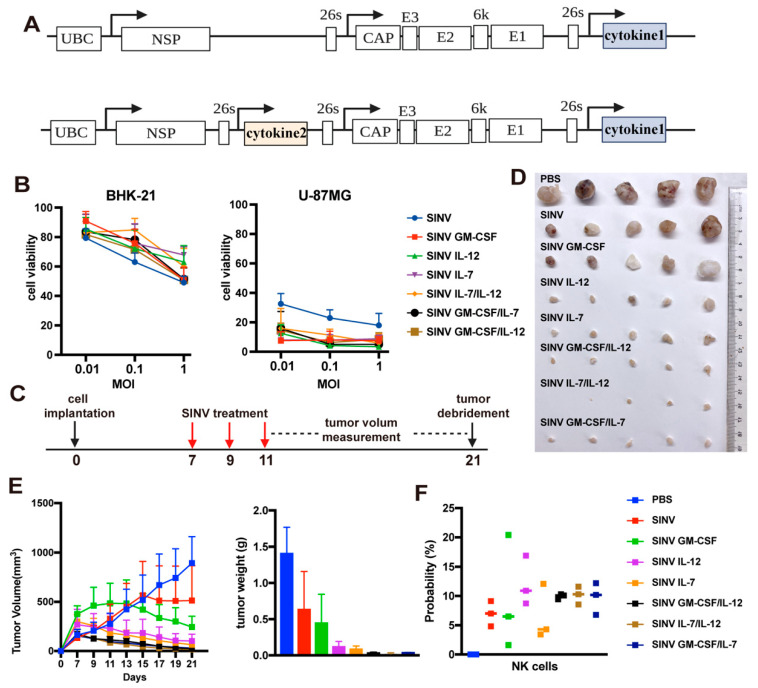
Cytokine arming improves therapeutic efficacy. The tumor-killing effect of 6 engineered SINVs carrying cytokines on U-87MG. (**A**) The schematic diagram of recombinant viral vectors (**B**) BHK-21 and U-87MG cells were infected with recombinant SINVs (MOI = 1, 0.1, 0.01) and at 48 h post-injection, the effect of each virus on the viability of both cells was counted (n = 6). (**C**) Treatment strategy. A total of 5 × 10^6^ U-87MG-Luc cells with 100 µL Matrigel were implanted into the thigh of mice. Recombinant SINVs were intratumorally injected into glioma tumors on days 7, 9, and 11 after cell implantation. (**D**) Representative tumor photo at 21 days post-treatment. (**E**) Volume–time plot of mouse tumors during treatment and comparison of tumor weights at 21 days after treatment (n = 5). (**F**) Comparison of NK cells within the tumor after 4 days of treatment. Error bars indicate S.E.M.; results are representatives of 3 independent experiments.

**Figure 5 cancers-15-04738-f005:**
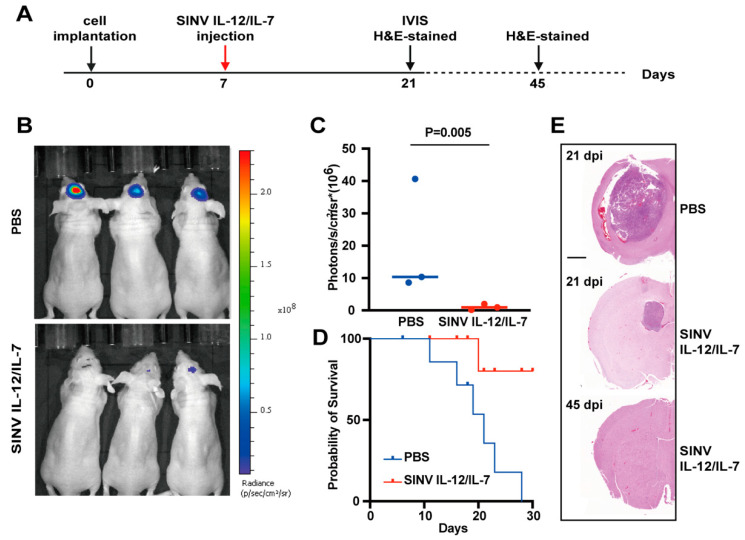
The performance of SINV IL-12/IL-7 in the U-87MG intracranial model. (**A**) Experimental timeline for SINV IL-12/IL-7 treatment in GBM model. (**B**) Luciferase imaging of U-87MG tumors 21 days after tumor implantation (n = 3). (**C**) The quantitative result of luciferase in (**B**) plot. Mean ± s.e.m., n = 3 mice per group, two-tailed unpaired *t*-test with Welch correction. (**D**) Survival curves of U-87MG-loaded mice treated with PBS, SINV IL-12/IL-7, respectively (n = 6). (**E**) H&E-stained coronal sections of U-87MG tumor-bearing mice brains treated at 21 days and 45 days post-cell implantation, respectively. Representative images of each group are presented. Scale bar, 1000 µm.

**Figure 6 cancers-15-04738-f006:**
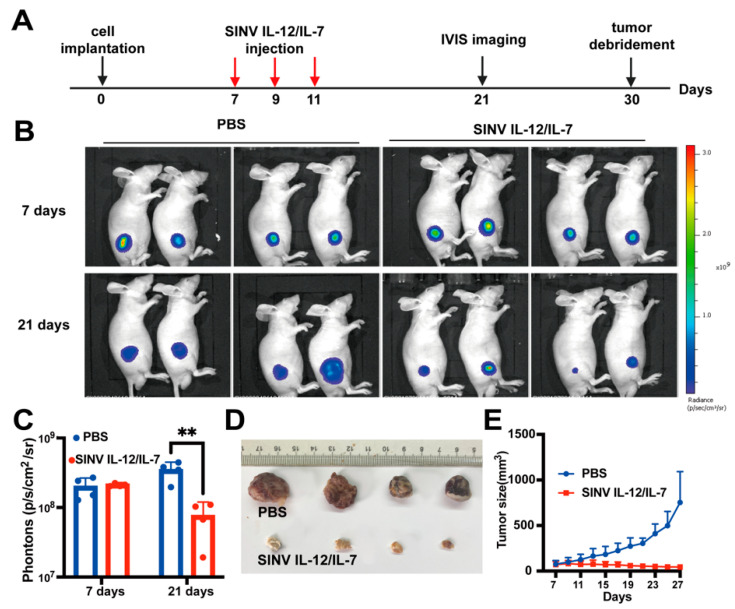
SINV carrying IL-12 and IL-7 can attack distal lesions. (**A**)Treatment options. 100 µL of SINV (1 × 10^7^ PFU) was injected into the tail vein of mice on days 7, 9, and 11 after cell transplantation. Tumors expressing luciferase were observed by IVIS on the day of administration and after administration. (**B**) Luciferase imaging of U-87MG tumors before and after treatment (n = 4). (**C**) Corresponding quantification of luciferase expression in (**B**). Mean ± s.e.m., n = 4 per group, two-tailed unpaired *t*-test, Welch’s correction. Quantitative results 21 days after treatment were statistically significant and are labeled with “**” in (**C**). (**D**) Comparison of tumors removed after treatment. (**E**) Volume–time plot of mouse tumors during treatment.

## Data Availability

The data that support the findings of this study are available from the corresponding author upon reasonable request.

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
