# Peer review of "Oncolytic Viral Therapy for Glioma by Recombinant Sindbis Virus"

_cancers, 2023, doi:10.3390/cancers15194738_

Round 1

Reviewer 1 Report

Recombinant SINV rescue information and characterisation is missing. SINV and recombinant SINV expressing cytokine replication kinetics are missing. Expression of the cytokines by the rSINV and quantifying them in cell culture supernatants in a dose-dependent response is required to characterise the viruses further. Sequence information of the cytokine is missing. Are they human or mouse origin?

It's crucial to functionally characterise IL7, IL12, and GM-CSF expressed by SINV; incubation of the cell culture supernatants infected with the PBMCs and IFNγ ELISpot is one way to do it.

U-87MG-Luc cell production authors have repeatedly transduced the cells with lentiviral supernatant to achieve stable Luciferase-expressing cells. The cells should be clonally sorted to generate cells.

Figure 1 authors claim that lymphocytes have decreased in the spleen 60 days post-virus treatment. Could authors emphasize why they chose this time point? The virus has already been cleared from the bloodstream long back. Immunocompetent C57BL/6 mice clear the virus from blood very fast, and no new information is given here.

Why did the authors use different virus concentrations 0.1 μL (1.0x10^5 PFU) of SINV-EGFP for safety were as injected with 3 μL (1x10^6 PFU) of SINV IL-12/IL-7?

SINV should be included in experiments 3.5 and 3.6 as a control

Figure 2 microscopy images are of poor quality; authors need to replace the figure with higher resolution and with a scale bar. Cell viability should be represented as mean with +/- standard deviation, not SEM.

Figure 6, IVIS for tumors is measured for systemic spread or tumor progress of unmeasurable areas like the brain, but IVIS measurements for subcutaneous tumors are not common. Why didn't the authors include classical tumor volume (mm3) as they did in figure 4?

Finally, if we compare the SINV-EGFP (1x10^6 PFU) treatment of U-87MG cells in Figure 3 and SINV IL-12/IL-7 (1.0x107 PFU) in Figure 6 are very similar. There is no additional benefit of IL-12/IL-7.

The virus dose was written 1.0x10^7 PFU in line 352 were as figure legend 1x10^8 PFU in line 362, and the authors should correct it.

Reviewer 2 Report

The study by Kangyixin et al., evaluated the oncolytic virus for efficacy and safety for the treatment of glioblastoma. The team investigated the efficacy of Sindbis virus treatment in different glioblastoma cell lines and in mouse models of subcutaneous and orthotopic GBM tumors. The study is well constructed, the results are well presented, and I would like to congratulate the authors on their very comprehensive work.

There are several issues that would require some clarification:

1. The introduction would benefit from a more detail presentation of Sindbis virus, including general characteristics, pathogenicity, general structure, where it is found, etc.

2. Materials and methods – Viruses and titration subchapter should include virus production (cell transfection methodology, virus purification, storage, etc.).

3. There is a discrepancy between the tumour growth pattern of tumours treated with virus alone as shown in Fig 3B and Fig 4D. Is there any explanation for this difference? Can the authors address this issue?

Other small issues are:

-       Sindbis virus is terminology throughout the text is not consistent (lower case or upper case).

-       Lines 34-36: Paragraph should be reformulated.

-       Line 55: Correct virus family name

-       Lines 84-94: Check the cell lines subchapter for typos and unnecessary spaces.

-       Line 98: Should be vector.

-       Line 114: The virus was  stored at -80C.

-       Line 130 – 131: correct the sentence, the cells are infected with virus not the other way around.

-       Line 148: experiments.. were  approved (insert were)

-       Line 197: Reformulate “we firstly..”

-       Line 205: remove “in mice” (repetition).

-       Lines 228-230: reformulate first sentence of figure legend.

-       Line 263: check for spaces

-       Line 274 and Figure 3… Figure panels are wrongly cited in text and the legend, should include a 3D.

-       Line 315: Figure symbol should be “E”.

-       Line 344: Uppercase for subchapter title.

-       Lines 389-390: heparane sulfate

-       Like 406: explain PKR abbreviation

The quality of English language is overall good. However, there are small issues that should be addressed, some of which I mentioned in my comments above. I recommend the authors to review the manuscript carefully and to seek native speaker input if possible.

Round 2

Reviewer 1 Report

I appreciate the authors address all the points.

But Figure 2 microscopy images are of poor quality; authors must replace the figure with higher magnification. They are still not clear. 
